

# Effects of exercise intervention on executive function in children with overweight and obesity: a systematic review and meta-analysis

Pengfei Wang[1], Ying Meng[1], Jinnian Tong[1] and Tiance Jiang[2]

[1] Department of Sports and Leisure, Dongshin University, Naju, Jeollanam-do, Republic of South Korea
[2] Physical Education Institute, Tomsk State University, Tomsk Oblast, Russia

## ABSTRACT

**Background**. Weight control in children depends on executive function. Previous studies have shown that exercise interventions can effectively improve children's executive function. However, the effects of these interventions on children with overweight and obesity remain unclear and require further investigation. This systematic review and meta-analysis was conducted to evaluate the effects of exercise interventions on executive function-related indicators in children with overweight and obesity.

**Methods**. Randomized controlled trials (RCTs) investigating the effects of exercise interventions on executive functions in children with overweight and obesity were included by searching PubMed, Web of Science, EMbase, Cochrane Library, ProQuest, Scopus, CNKI, China Wanfang, and VIP databases. The quality of the included studies was assessed using the Cochrane risk of bias assessment tool. RevMan 5.4 software was used for effect size pooling, forest plot creation, and subgroup analyses. Stata 16.0 software was employed for publication bias testing and sensitivity analysis. The evidence levels of the results were evaluated using the GRADEpro tool.

**Results**. This meta-analysis included a total of 13 studies. The results indicate that exercise interventions may help improve executive functions in children with overweight and obesity. Specifically, inhibitory control (standardized mean (SMD) $= -0.59$, 95% confidence interval (CI) $[-0.89$ to $-0.29]$, $Z = 3.85$, $P < 0.001$) and cognitive flexibility (SMD $= -0.54$, 95% CI $[-1.06$ to $-0.01]$, $Z = 2.01$, $P < 0.05$) showed moderate effect sizes. Working memory exhibited a smaller effect size (SMD $= 0.40$, 95% CI $[-0.69$ to $-0.10]$, $Z = 2.61$, $P < 0.01$), while attention did not show significant improvement (SMD $= 0.13$, 95% CI $[-0.39$ to $0.65]$, $Z = 0.50$, $P > 0.05$).

**Conclusion**. The results of this meta-analysis indicate that exercise interventions have significant benefits for inhibitory control, working memory, and cognitive flexibility in children with overweight and obesity, but the impact on attention is not significant. Moreover, the effects of inhibitory control interventions are influenced by exercise duration, exercise intensity, exercise type, and age.

Corresponding author
Tiance Jiang, 17624003833@163.com

## INTRODUCTION

Obesity and overweight in children have become a global health problem (*Wang et al., 2017*). In recent decades, the prevalence of overweight and obesity among children has increased annually in many countries (*Hong et al., 2023*). Data show that by 2022, the total number of children and adolescents with obesity worldwide will reach 159 million, and the obesity rate will be approximately four times that of 1990 (*NCD-RisC, 2024*). The incidence of chronic diseases such as hypertension and diabetes in individuals with overweight and obesity is significantly higher than that in normal-weight individuals (*Guo et al., 2019*; *Huang, Li & Chen, 2019*). In addition, being overweight and obese are associated with poor cognitive function, especially executive function (*Likhitweerawong et al., 2022*). Executive function refers to an advanced cognitive process that regulates various basic cognitive processes when individuals complete complex tasks. Executive function has a multidimensional structure that mainly includes three core components: inhibition control, working memory, and cognitive flexibility (*Doebel, 2020*), which can promote individual cognitive and psychosocial development and are conducive to academic and life success (*Diamond, 2020*). Studies have found that insufficient executive function in early childhood may predict whether children develop obesity. Poor executive function reduces an individual's self-control and makes them more likely to pay too much attention to food-related cues, leading to overeating and eventually obesity (*Pieper & Laugero, 2013*; *Riggs et al., 2010*). A longitudinal study also found that low executive function at age four is closely associated with high body mass index (BMI) at age six, suggesting that a decline in executive function may be one of the reasons leading to obesity (*Guxens et al., 2009*). Therefore, solving the problem of executive functioning in children with overweight and obesity may be an effective means of improving overweight and obesity.

There is evidence that exercise interventions have a good effect on improving executive function. The mechanisms by which exercise interventions affect executive function may mainly be based on the following two aspects. First is brain plasticity: the functions and structures of the brain are continuously modified and reorganized in response to changes within the body and the external environment (*Green & Bavelier, 2008*). This is usually manifested as an increase in the volume of total gray matter and cerebellar gray matter, and an improvement in the efficiency of overall neural circuits in the brain to enhance children's executive function (*Burdette et al., 2010*; *Davis et al., 2011*; *Xiong et al., 2018*). This mechanism runs throughout the entire human lifespan, affecting individuals from childhood to old age (*Green & Bavelier, 2008*). Second, based on neurobiological hypotheses, engaging in physical activities promotes changes in the central nervous system. This is mainly manifested by the generation of new neurons in the hippocampus, the formation of blood vessels in the brain, and an increase in gray matter volume in brain regions related to learning and memory (*Dishman et al., 2006*; *Singh et al., 2019*). Some animal studies (*Rhodes et al., 2003*; *Vaynman, Ying & Gomez-Pinilla, 2004*) and correlational and interventional neuroimaging studies in the elderly support this hypothesis (*Colcombe et al., 2006*; *Reiter et al., 2015*; *Ruscheweyh et al., 2011*).

In recent years, a large number of studies have confirmed the benefits of exercise interventions on children's executive functions (*Amatriain-Fernández, Ezquerro García-Noblejas & Budde, 2021*; *Chen et al., 2023*; *Contreras-Osorio et al., 2021*; *Feng et al., 2023*; *Liu et al., 2020*; *Song et al., 2023*; *Wang et al., 2023*; *Wu et al., 2023*; *Xue, Yang & Huang, 2019*; *Zang et al., 2024*). However, the populations involved have primarily been normal-weight children, with relatively few studies focusing on children with overweight and obesity. *Lin et al. (2024)*'s research explored the impact of chronic exercise interventions on the executive functions of obese children. However, this study did not delve into whether there are differences in the effects of exercise interventions among children of different age groups. Additionally, the exercise interventions included in the study were all long-term exercises and did not incorporate acute exercises. Whether there are differences in the intervention effects between these two types of exercise interventions remains to be investigated.

Considering the complexity of the relationship between exercise intervention and executive function in children with overweight and obesity, this study objectively evaluated existing research results through a meta-analysis, providing evidence-based guidance for the study of exercise intervention in the field of executive function in children with overweight and obesity.

## MATERIALS & METHODS

### Protocol and guidance

This study was reported according to the Preferred Reporting Project for Systematic Review and Meta-Analysis (PRISMA) guidelines (*Liberati et al., 2009*). This evaluation agreement was registered with INPLASY (INPLASY202470043).

### Inclusion criteria

The trial type is a randomized controlled trials. Study subjects are children with overweight and obesity (under 18 years old) who meet the WHO criteria for overweight and obesity (*Du & Ma, 2006*). Our research focuses on exercise interventions, and studies will be considered for inclusion if they compare any form of exercise intervention with routine activities, non-exercise interventions, or no intervention. Outcome measures must include at least one indicator that can be used to calculate the effect size of executive functions (inhibitory control, working memory, cognitive flexibility, attention).

### Exclusion criteria

The trial type is a non-randomized controlled trials. Study subjects are non-overweight or non-obese children. The intervention in the experimental group is not an exercise intervention, or all participants undergo exercise interventions. The outcome measures in the study do not meet the criteria or cannot be extracted. The study language is not Chinese or English.

### Outcomes

Outcome measures included inhibitory control, working memory, cognitive flexibility, and attention.
## Search strategy

Two independent researchers (PW and TJ) searched multiple databases: PubMed, Web of Science, Embase, Cochrane Library, ProQuest, Scopus, CNKI, China Wanfang, and VIP from their creation until July 1, 2024. If the two researchers encounter disagreements during the retrieval process, a third researcher (JT) will join the discussion. We did not restrict the languages used. The following keywords were used: exercise intervention (exercise OR aerobic exercise OR aerobic training OR physical activity OR physical exercise OR taijiquan OR dancing OR cycling OR swimming OR jogging OR walking OR vigorous walking OR treadmill OR qi gong OR ba dual jin OR stair climbing OR isometric OR acute exercise OR endurance training OR sports OR athletics OR resistance training OR strength training OR resistance exercise OR strength exercise OR weight training OR weight-lifting strengthening program OR weight lifting exercise program OR high-intensity interval training OR high-intensity intermittent exercise OR sprint interval training OR hint OR yoga OR circuit-based exercise OR plyometric exercise OR plyometric training OR stretch shortening cycle exercise OR stretch-shortening exercise OR plyometric drills OR cycle exercise OR ball OR soccer ball OR soccer OR football OR basketball OR ping-pong OR badminton OR tennis OR baseball OR volleyball OR softball OR racquet sport OR racket sport OR lacrosse OR racquetball OR exercise OR dancing), executive function (executive function OR executive control OR cognitive function OR cognitive performance OR inhibitory control OR shifting OR working memory OR refresh function OR cognitive flexibility OR planning OR cognitive function OR reasoning OR problem solving OR updating OR inhibition OR attention), and study population (child OR children), and study population characteristics (overweight OR obesity OR obese OR excess weight).

## Study selection

After removing duplicates, two independent researchers (PW and TJ) initially screened the papers based on titles and abstracts and then further screened the papers by reading the full text. If there was a disagreement between the two independent researchers regarding the selection of articles, a third researcher (JT) participated in the decision.

## Data collection process

Two researchers independently extracted information from the literature for analysis. The extracted information included (1) basic information: author, publication year, and study design; (2) participant characteristics: age, BMI, and sample size; and (3) exercise intervention program: intervention cycle, intervention frequency, duration of single intervention, exercise intensity, and exercise type. The type of exercise intervention was divided into open and closed motor skills according to the main motor skill characteristics in sports. A characteristic of open motor skills is that athletes cannot decide in advance how to perform the next movement before performing it, but must decide according to the stimuli given by the external environment at that time, such as in ping-pong sports. Closed motor skills refer to the relatively stable stimulation of the external environment, and athletes can decide what to do in advance before performing the next action, such as in running (*Shi & Wang, 2014*). Exercise intensity can be divided into moderate-intensity and

high-intensity exercises according to the center rate of the exercise process. (4) Inhibition control outcome indicators were represented by performance in Stroop, Flanker, and food-cue-related Stroop tests; working memory outcomes were represented by performance in 1-back and N-back tests. Cognitive flexibility outcome indices were represented by the performance on the more-odd shifting and Wisconsin Card Sorting Test (WCST) tests. The index of attention was represented by the d2-R test and the performance of the "attention" in cognitive assessment system (CAS).

## Quality evaluation of literature

According to the Cochrane Collaboration's risk-of-bias assessment tool, two researchers (PW and TJ) used Revman 5.4 software to assess the risk of bias in the included studies (*Higgins, 2008*). Two researchers evaluated the following seven items: (1) random sequence generation; (2) allocation concealment; (3) blinding of participants and personnel; (4) blinding of outcome assessment; (5) incomplete outcome data; (6) selective reporting; and (7) other bias. The study's risk of bias was classified as "low", "high", and "unclear".

## Statistical analysis

Meta-analysis was performed using RevMan 5.4 and Stata 16.0 software. All the included outcome indicators were continuous variables, but due to different assessment tools, we used the standardized mean (SMD) as the effect size and calculated the 95% confidence interval (95% CI) of the effect size. An effect size $< 0.2$ indicated a negligible effect, $0.20-0.49$ was a small effect, $0.50-0.79$ was a moderate effect, and $\geq 0.8$ was a large effect (*Cohen, 2013*). The Q test and I statistic were used to investigate the inter-study heterogeneity. $P \geq 0.1$ or $I^2 \leq 50\%$ indicated no inter-study heterogeneity, and the combined effect size of fixed effects was used; conversely, the combined effect size of random effects was used, and further subgroup studies were used to investigate the source of heterogeneity if necessary. Additionally, heterogeneity among studies was quantified using the $I^2$ statistic. The levels of heterogeneity were classified as small ($I^2 \leq 25\%$), moderate ($25\% < I^2 \leq 50\%$), substantial ($50\% < I^2 \leq 75\%$), and considerable ($I^2 > 75\%$) (*Higgins et al., 2003*). A sensitivity analysis was used to examine the extent to which the results of each study affected the combined effect size. This study assessed publication bias using intuitive funnel plots and quantitative Egger's and Begg's tests. A $P$-value of 0.05 was used as the critical threshold; a $P > 0.05$ indicates no publication bias, while a $P < 0.05$ suggests the presence of publication bias.

## Subgroup analyses

We conducted the following subgroup analyses stratified by exercise duration (acute exercise *versus* chronic exercise), exercise type (open motor skills *versus* closed motor skills), exercise intensity (moderate *versus* high), and age ($\geq 12$ years *versus* $< 12$ years) to examine the interactions.

## Sensitivity analyses

A sensitivity analysis was conducted by randomly deleting one of the studies included in each index to observe whether the deleted study would exert a large change in the overall effect size.

## Quality evaluation of outcome evidence

We use GRADEpro software to assess the quality of evidence for outcomes. The quality of evidence is divided into high, moderate, low, and very low. High: We are very confident that the results of the systematic review are truly reliable. Moderate: The results of the systematic review may be close to the true value, but there is a significant possibility that they differ. Low: The results of the systematic review are very likely not the same as the true value. Very low: We have almost no confidence in the results of the systematic review; the results are likely different from the true value.

# RESULTS

## Results of literature screening and characteristics of the included studies

We initially identified 6,151 records, of which 49 were selected after reading the full text. After excluding studies with no executive function (eight), non-interventional studies (18), studies without exercise intervention (three), replicates of experiments (1), studies with all participants receiving exercise intervention (three), and studies on children without overweight or obesity (three), 13 eligible studies were included in the final meta-analysis (*Chen et al., 2016*; *Chou et al., 2023*; *Davis et al., 2007*; *Davis et al., 2011*; *Gallotta et al., 2015*; *Krafft et al., 2014*; *Liu et al., 2018*; *Xiang et al., 2019*; *Xie, 2020*; *Zhang et al., 2020*; *Zhang et al., 2022*; *Zhang, 2019*; *Zhu, 2022*). An overview of the screening process is shown in Fig. 1.

Of the 13 studies included, with a total of 963 participants, published between 2007 and 2023, five recruited overweight children, five recruited children with obesity, and three recruited both children with overweight and children with obesity, ranging in age from 8 to 15 years. All included studies reported all or part of the exercise intervention variables, including the intervention methods, plan, intensity, and type, as shown in Table 1.

## Quality evaluation of included literature

Four studies had a low risk of bias in random sequence generation, while the others were unclear. Two studies had a low risk of bias in allocation concealment, with the rest being unclear. Since all studies included in this research were exercise intervention studies, the possibility of unblinding was high; therefore, all studies were at high risk concerning the blinding of participants and personnel. Two studies had a low risk of bias in the blinding of outcome assessment, while the others were unclear. Twelve studies had a low risk of bias regarding the incomplete outcome data, and the remaining one had a high risk. All thirteen studies were at low risk of bias in terms of selective reporting and other bias. Details of the risk of bias evaluation are shown in Fig. 2.

## Inhibitory control

Among the 13 studies, a total of nine studies elucidated the effect of exercise intervention on inhibitory control in children with overweight and obesity, involving 528 children diagnosed with overweight or obesity. As shown in Fig. 3A, the overall effect indicates that, compared to the control group, exercise intervention has a medium effect size

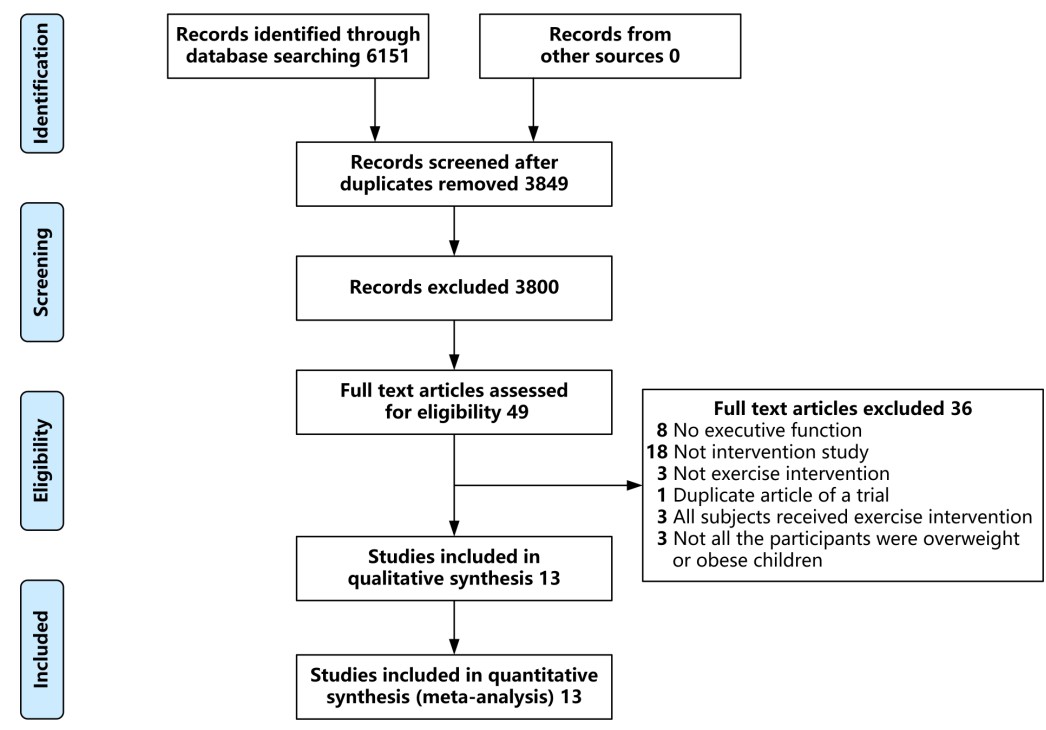

**Figure 1  Literature screening process.**

(SMD $= -0.59$, 95% CI [$-0.89$ to $-0.29$], $Z = 3.85$, $P < 0.001$). This effect exhibits significant heterogeneity ($I^2 = 62\%$, $P < 0.05$).

Table 2 shows the results of the subgroup analyses. Using exercise duration as a moderating variable for subgroup analysis, the results showed that chronic exercise had a large intervention effect on inhibitory control (SMD $= -0.85$, 95% CI [$-1.29$ to $-0.41$], $Z = 3.76$, $P < 0.001$), while acute exercise had a small to medium intervention effect (SMD $= -0.29$, 95% CI [$-0.55$ to $-0.03$], $Z = 2.23$, $P < 0.05$). Using exercise intensity as a moderating variable for subgroup analysis, the results showed that both high-intensity (SMD $= -0.63$, 95% CI [$-1.03$ to $-0.23$], $Z = 3.05$, $P < 0.01$) and moderate-intensity (SMD $= -0.64$, 95% CI [$-1.24$ to $-0.05$], $Z = 2.11$, $P < 0.05$) exercise interventions had moderate effects on inhibitory control. Using exercise type as a moderating variable for subgroup analysis, the results showed that open motor skills exercise (SMD $= -1.11$, 95% CI [$-1.72$ to $-0.50$], $Z = 3.55$, $P < 0.001$) had a large intervention effect on inhibitory control, while closed motor skills exercise had a small to medium intervention effect (SMD $= -0.40$, 95% CI [$-0.65$ to $-0.15$], $Z = 3.10$, $P < 0.01$). Using age as a moderating variable for subgroup analysis, the results showed that exercise interventions had a small to medium effect on children aged 12 and above (SMD $= -0.43$, 95% CI [$-0.76$ to $-0.10$], $Z = 2.56$, $P < 0.05$), and a moderate effect on children under 12 years old (SMD $= -0.69$, 95% CI [$-1.14$ to $-0.25$], $Z = 3.08$, $P < 0.01$).
**Table 1  Characteristics of included literature.**

| Included studies | Study design | Sample size | Average age or age range | Average BMI | Intervention methods | Intervention plan | Intervention intensity | Intervention type | Outcome measured |
|---|---|---|---|---|---|---|---|---|---|
| Zhang et al. (2020) | RCT | 38 | 14.63 ± 0.49 | 27.91 ± 3.26 | EG: Rope skipping<br>CG: Reading | 30 min* once | HR: 140–150 times/min | Closed | ①② |
| Krafft et al. (2014) | RCT | 50 | 9.80 ± 0.85 | Not reported | EG: Sports games + rope skipping<br>CG: Sedentariness | 40 min, 5 weekly, 32 weeks | Average HR: 161 times/min | Closed | ③ |
| Chou et al. (2023) | RCT | 108 | 11.23 ± 0.61 | 20.76 ± 0.68 | EG: Competitive team game<br>CG: Sedentariness | 40 min/20min, 5 weekly, 10 weeks | HR>150 times/min | Open | ① |
| Xiang et al. (2019) | RCT | 44 | 12.89 ± 1.51 | 29.33 ± 3.51 | EG: Aerobics, ball games, resistance exercise, etc<br>CG: No motion intervention | 300 min, 6 weekly, 6 weeks | Unclear | Closed | ① |
| Chen et al. (2016) | RCT | 60 | 12.74 ± 0.73 | 28.89 ± 3.50 | EG: Brisk walking, skipping rope, climbing stairs, dancing, etc<br>CG: health education courses | 40 min, 4 weekly, 12 weeks | 60%–70% HRmax | Closed | ④ |
| Zhang et al. (2022) | RCT | 72 | 11.56 ± 1.03 | 26.02 ± 1.05 | EG1: High intensity interval treadmill exercise<br>EG2: High intensity continuous skipping exercise<br>CG: Watch cartoons | 30 min*once | EG1: 85–95% HRmax | Closed | ① |
| Davis et al. (2007) | RCT | 97 | 9.20 ± 0.83 | 25.80 ± 4.00 | EG: Running games, jump rope, improved basketball and soccer<br>CG: Daily activities | 40 min/20min, 5 weekly, 15 weeks | HR>150 times/min | Open+Closed | ⑤ |
| Davis et al. (2011) | RCT | 171 | 9.30 ± 1.00 | 26.00 ± 4.60 | EG: Running games, jump rope, improved basketball and soccer<br>CG: No motion intervention | 40 min/20min, 5 weekly, 13 weeks | HR>150 times/min | Open+Closed | ⑤ |
| Gallotta et al. (2015) | RCT | 53 | 8∼11 | 25.58 ± 2.41 | EG1: Coordination training<br>EG2: Traditional training<br>CG: No motion intervention | 60 min, 2 weekly, 20 weeks | 5<RPE<8 | EG1: Open<br>EG2: Closed | ⑥ |
| Liu et al. (2018) | RCT | 70 | 14.06 ± 0.83 | 27.97 ± 3.14 | EG: Skipping<br>CG: Await | 75 min, 2 weekly, 12 weeks | HR: 150–170 times/min | Closed | ①② |
| Xie (2020) | RCT | 80 | 9.76 ± 0.66 | 21.62 ± 2.10 | EG: Jump rope + basketball<br>CG: Classroom study | 40 min/30 min/20 min*once | HR: 130–150 times/min | Open | ③⑦⑧ |
| Zhu (2022) | RCT | 40 | 10.65 ± 0.80 | 24.67 ± 2.18 | EG: Basketball<br>CG: Routine activity | 40 min, 3 weekly, 8 weeks | Average HR: 135 times/min | Open | ③⑧⑨ |
| Zhang (2019) | RCT | 80 | 10.35 ± 0.61 | 21.42 ± 0.91 | EG: Jump rope + basketball<br>CG: Self-study | 40 min/20 min* once | HR: 130–150 times/min | Unclear | ③⑦⑧ |

**Notes.**
HR, Heart Rate; HRmax, Heart Rate maximum; EG, Experimental Group; CG, Control Group; RCT, Randomized Controlled Trial; RPE, Rating of Perceived Exertion; ①, Stroop test; ②, related Stroop test; ③, Flanker test; ④, Wisconsin Card Sorting Test; ⑤, Cognitive Assessment System; ⑥, d2-R test; ⑦, N-Back; ⑧, More-odd shifting; ⑨, 1-Back.

## Working memory

Among the 13 studies, a total of three studies elucidated the effect of exercise intervention on working memory in children with overweight and obesity, involving 200 children diagnosed with overweight or obesity. As shown in Fig. 3B, the overall effect indicates that, compared to the control group, exercise intervention has a small effect size (SMD = −0.40,

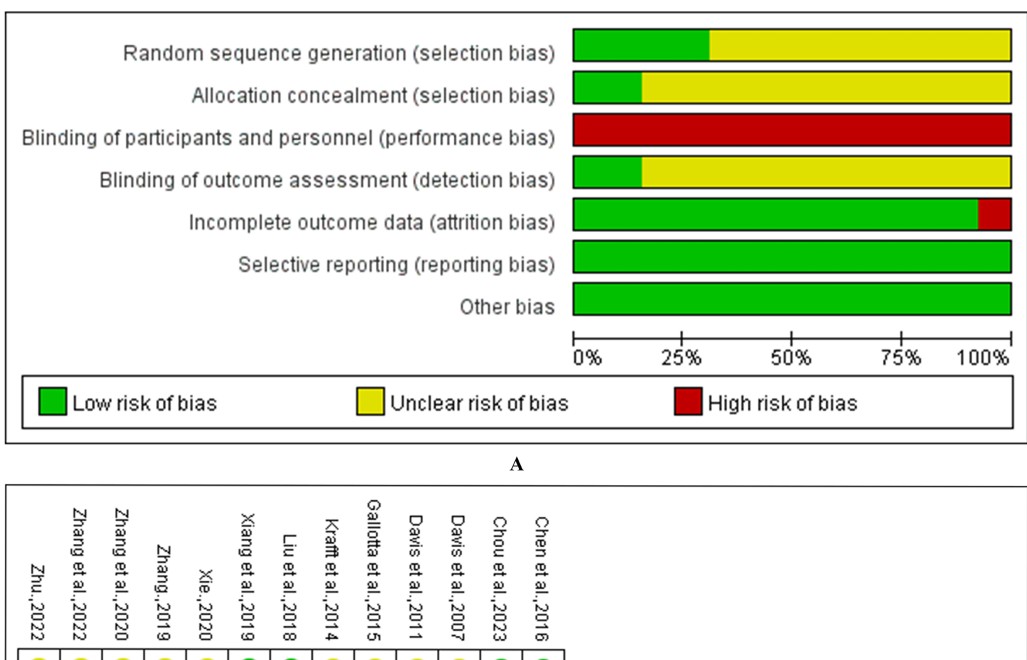

**Figure 2  Risk of bias assessment for the included studies.** (A) methodological quality of included studies; (B) distribution of the methodological quality of included studies.

95% CI: $-0.69$ to $-0.10$, $Z = 2.61$, $P < 0.01$). This effect exhibits moderate heterogeneity ($I^2 = 50\%$, $P > 0.05$).

### Cognitive flexibility

Among the 13 studies, a total of four studies elucidated the effect of exercise intervention on cognitive flexibility in children with overweight and obesity, involving 250 children diagnosed with overweight or obesity. As shown in Fig. 3C, the overall effect indicates that, compared to the control group, exercise intervention has a medium effect size (SMD $= -0.54$, 95% CI [$-1.06$ to $-0.01$], $Z = 2.01$, $P < 0.05$). This effect exhibits significant heterogeneity ($I^2 = 73\%$, $P < 0.05$).

### Attention

Among the 13 studies, a total of three studies elucidated the effect of exercise intervention on attention in children with overweight and obesity, involving 308 children diagnosed

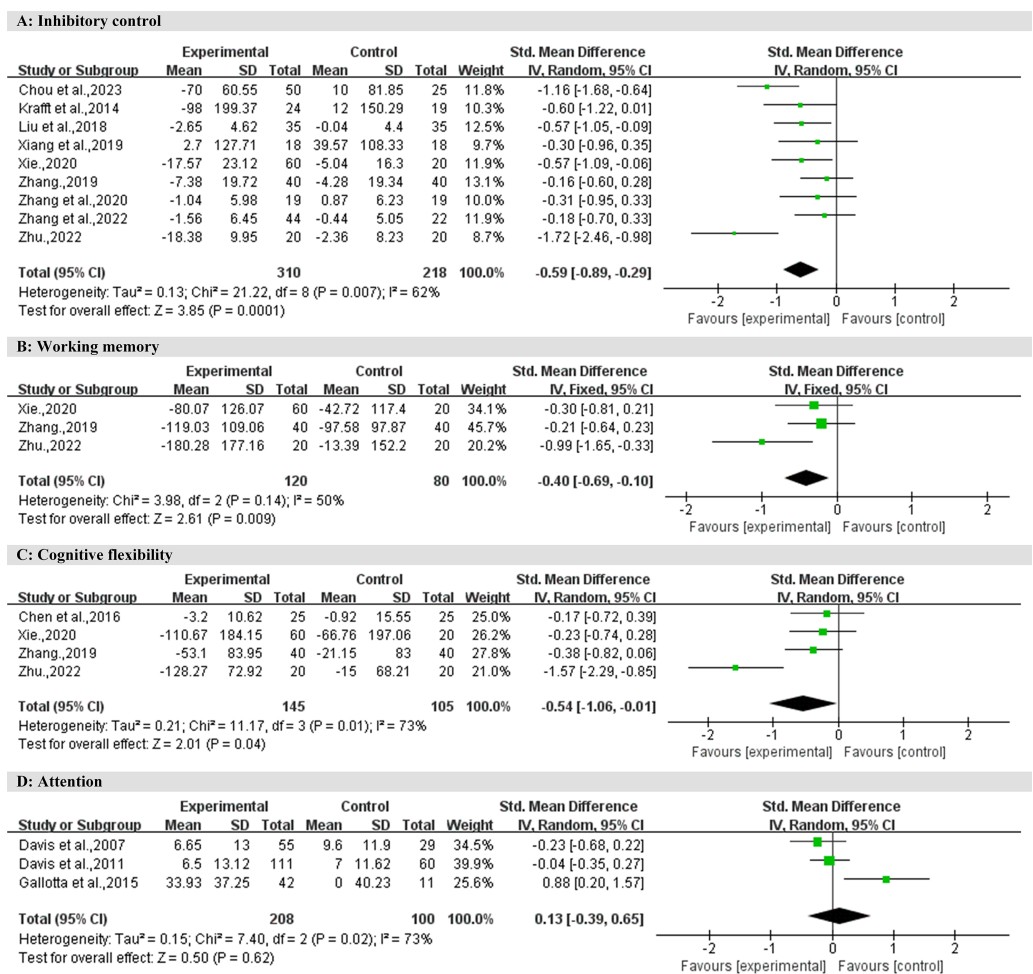

**Figure 3** Forest plot for meta-analysis regarding the effect of exercise interventions on different executive function domains.

with overweight or obesity. As shown in Fig. 3D, the overall effect indicates that, compared to the control group, the effect size of exercise intervention is negligible (SMD = 0.13, 95% CI [−0.39 to 0.65], Z = 0.50, P > 0.05). This effect exhibits significant heterogeneity ($I^2$ = 73%, P < 0.05).

## Publication bias test

The funnel plot for inhibitory control did not show asymmetry; Egger's test (P = 0.226 > 0.05) and Begg's test (P = 0.175 > 0.05) did not detect publication bias. The funnel plot for working memory did not show asymmetry; Egger's test (P = 0.139 > 0.05) and Begg's test (P = 0.296 > 0.05) did not detect publication bias. The funnel plot for cognitive flexibility did not show asymmetry; Egger's test (P = 0.230 > 0.05) and Begg's test (P = 1.000 > 0.05) did not detect publication bias. The funnel plot for attention did not show asymmetry; Egger's test (P = 0.485 > 0.05) and Begg's test (P = 1.000 > 0.05) did not detect publication bias, as shown in Fig. 4.

**Table 2  Subgroup analysis of the effect of exercise interventions on inhibitory control in children with overweight and obesity.**

| Adjustable variable | Subcategory | Number of studies included | Effect model | Results of heterogeneity testing | | Meta-analysis result | | |
|---|---|---|---|---|---|---|---|---|
| | | | | *P* | *I²* | SMD | 95% CI | *P* |
| Exercise duration | Acute exercise | 4 | Random | 0.64 | 0% | −0.29 | −0.55, −0.03 | 0.03* |
| | Chronic exercise | 5 | | 0.02 | 65% | −0.85 | −1.29, −0.41 | <0.001*** |
| Exercise intensity | Moderate-intensity exercise | 4 | Random | 0.004 | 77% | −0.64 | −1.24, −0.05 | 0.03* |
| | High-intensity exercise | 4 | | 0.07 | 57% | −0.63 | −1.03, −0.23 | 0.002** |
| Exercise type | Closed motor skill | 5 | Random | 0.78 | 0% | −0.4 | −0.65, −0.15 | 0.002** |
| | Open motor skill | 3 | | 0.04 | 70% | −1.11 | −1.72, −0.50 | <0.001*** |
| Age | ≥12 years old | 3 | Random | 0.73 | 0% | −0.43 | −0.76, −0.10 | 0.01* |
| | <12 years old | 6 | | 0.001 | 75% | −0.69 | −1.14, −0.25 | 0.002** |

**Notes.**
*\*P*-value less than 0.05.
*\*\*P*-value less than 0.01.
*\*\*\*P*-value less than 0.001.

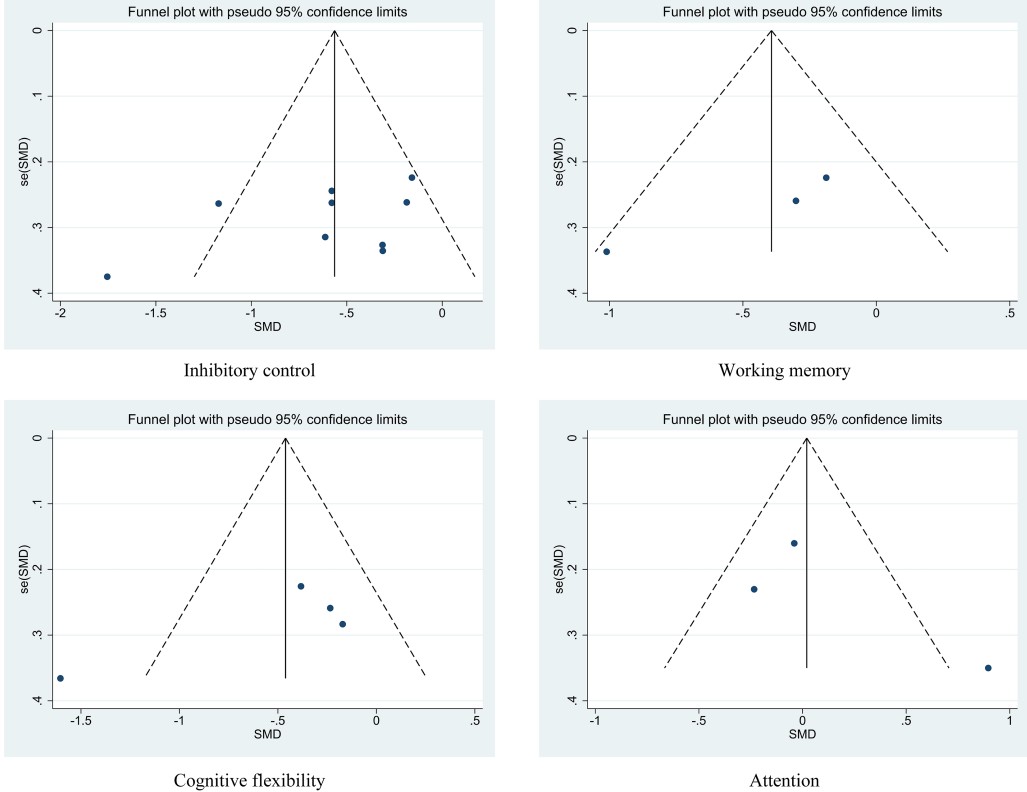

**Figure 4  Funnel plot for visual inspection of publication bias.**

## Sensitivity analyses

Using a stepwise exclusion method, sensitivity analyses were performed on inhibitory control, working memory, cognitive flexibility, and attention, respectively. The results

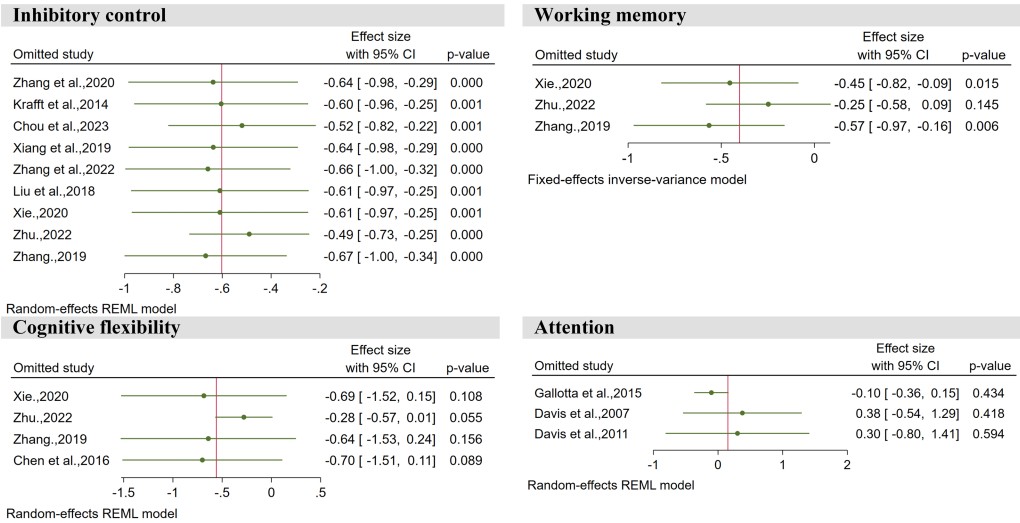

**Figure 5 Sensitivity analysis plot.**

showed that for each indicator, after excluding any one of the studies, the combined effect size results of the remaining studies remained stable and did not significantly affect the overall conclusions of the research, as shown in Fig. 5.

## Evaluation of outcome evidence

Using GRADEpro to evaluate each outcome, the result for inhibitory control was of moderate quality, while the results for working memory, cognitive flexibility, and attention were of low quality, as shown in Table 3.

## DISCUSSION

The results of this study show that exercise interventions can improve inhibitory control, working memory, and cognitive flexibility in children with overweight and obesity, which is similar to the results of several previous systematic reviews (*Lin et al., 2024*; *Sun et al., 2021*; *Zhao et al., 2024*). There is an association between body mass index (BMI) and certain specific parts of the brain structure. When the body is in an obese state (BMI > 30), there is a certain degree of atrophy in the frontal lobe, hippocampus, anterior cingulate gyrus, and thalamus (*Raji et al., 2010*). When the body is in an overweight state (BMI: 25~30), there is a certain degree of atrophy in the corona radiata of the white matter and basal ganglia (*Raji et al., 2010*). Exercise interventions can enhance their executive functions through mechanisms such as improving brain structures related to executive functions, regulating neurotransmitter levels, and enhancing neuroplasticity, thereby ultimately achieving the purpose of weight loss (*Dishman et al., 2006*; *Green & Bavelier, 2008*; *Voss et al., 2011*). Of course, exercise itself will also directly bring weight loss benefits. In addition, some studies have shown that when individuals with obesity successfully lose weight through bariatric surgery, their executive functions also improve (*Gurnani et al., 2022*). This may suggest that there is a complex bidirectional relationship between obesity and executive

**Table 3 Level of evidence for outcome indicators.**

| Outcome | Study design | Evaluation of evidence quality level | | | | | Quality of evidence |
|---|---|---|---|---|---|---|---|
| | | Risk of bias | Inconsistency | Indirectness | Imprecision | Publication bias | |
| Inhibitory control | RCT | serious | no | no | no | no | Moderate |
| Working memory | RCT | serious | no | no | serious | no | Low |
| Cognitive flexibility | RCT | serious | no | no | serious | no | Low |
| Attention | RCT | serious | no | no | serious | no | Low |

**Notes.**
RCT, Randomized Control Trial; serious, downgrade rating by one level; no, no downgrade.

function impairment (*Lowe, Reichelt & Hall, 2019*; *Martí-Nicolovius, 2022*). However, there is currently no definite conclusion on the order in which the two occur, which can be a direction for future research. But it is currently certain that exercise interventions can have a positive impact on the executive functions of children with overweight and obesity.

We found that exercise interventions did not improve the attention of children with overweight and obesity, which is inconsistent with previous findings. A systematic review published in 2017 analyzed 31 experiments with 4,953 children and found that acute physical intervention improved the attention of healthy children to some extent (Hedges' $g = 0.43$; 95% CI $= 0.09, 0.77$; $p = 0.013$) (*De Greeff et al., 2018*). The reason for this may be that the participants were children with overweight and obesity, whose attention was significantly lower than that of normal children and was more difficult to improve (*Liang et al., 2014*; *Reinert, Po'e & Barkin, 2013*). In addition, improvement in children's attention has a sensitive period, starting from 4–5 years old, when children's intentional attention begins to develop (*Wu & Zhang, 2020*). Children at this stage are more easily stimulated by the external environment and can more easily improve. They also achieve better improvement effects. After the age of 7 years, children's attention levels gradually stabilize and they are less susceptible to outside interference, and the effect of sports intervention will also be affected to some extent (*Wu & Zhang, 2020*). Notably, the choice of exercise intensity significantly affects the improvement in children's attention. Several studies have found that moderate-intensity exercise interventions are most effective in improving children's attention, while high-intensity exercise may, in some cases, have a negative impact on children's attention (*Budde et al., 2008*; *Chang et al., 2012*; *Hillman et al., 2009*). Therefore, if we want to use exercise intervention to improve the attention of children with overweight and obesity, we should try to intervene before the age of seven years and control the exercise intensity at a moderate level.

According to the subgroup analysis by exercise duration, both acute and chronic exercise significantly improved inhibitory control in children with overweight and obesity, but the intervention effect of chronic exercise was significantly better (effect size (ES) $= -0.85$). This may be due to the longer intervention cycle of chronic exercise, which is usually eight weeks or more, and some may last for several years. The neurophysiological mechanism suggests that chronic exercise can cause structural changes in the human brain, specifically, in brain regions related to learning and memory; promote the regulation of growth factors

in the brain; and cause neurodevelopment (*Carey, Bhatt & Nagpal, 2005*). On the other hand, acute exercise is a brief or one-time physical activity, and although it also has a positive impact on the inhibitory control of children with overweight and obesity, its intervention effect is significantly lower than that of chronic exercise (ES = −0.29), and the intervention effect of acute exercise also lasts for a shorter period of time, usually around 30 minutes (*Lambrick et al., 2016*). Therefore, if we want to use an exercise intervention to improve the executive function of children with overweight and obesity, a chronic exercise intervention is a good choice under the conditions that it is allowed.

According to the subgroup analysis by exercise type, exercise based on both open and closed motor skills significantly improved the inhibitory control of children with overweight and obesity, but intervention based on open motor skills has a better effect (ES = −1.11), which is consistent with the results of several similar studies (*Feng et al., 2023*; *Qiu, Zhai & Chen, 2024*). The reason may be because projects based on open motor skills have more complex environments and higher cognitive involvement, which not only promotes physiological arousal but also activates cognitively related neural networks to a greater extent, thereby improving executive function. From the perspective of cognitive influence, higher cognitive involvement is required when performing complex exercises, which involves more neural circuits and enhances the activity of the prefrontal cortex, thereby promoting the improvement of executive function (*Xia et al., 2018*). Although closed motor skills can also positively affect the inhibitory control of children with overweight and obesity (ES = −0.40), the external environment and technical actions to be completed are always the same, and participants can control their movements solely by proprioception and do not need to mobilize too many cognitive resources. This may explain why the intervention effect of closed motor skills was not as good as that of open motor skills. Additionally, recent literature has proposed two teaching models: linear pedagogy and nonlinear pedagogy (*Rudd et al., 2020*). Linear pedagogy, as a relatively traditional teaching model, emphasizes the systematic and gradual acquisition of technical skills (*Rudd et al., 2020*). It employs continuous or segmented training and stabilizes movements through repetitive practice (*Rudd et al., 2020*). This approach is more suitable for the mastery of closed motor skills, because closed motor skills are performed in stable and predictable environments. During repetitive practice, students need to inhibit incorrect movement patterns and focus on correct technical execution, which helps enhance their inhibitory control ability (*Diamond, 2013*). On the other hand, nonlinear pedagogy is a more flexible teaching model. Teachers adjust teaching strategies based on changes in tasks and the environment to promote students' autonomous learning and creative problem-solving (*Rudd et al., 2020*). This method emphasizes situational diversity and the dynamic nature of the learning process, making it suitable for the mastery of open motor skills, because open motor skills are characterized by dynamism and unpredictability (*Rudd et al., 2020*). When students face different challenges, they need to flexibly adjust strategies, inhibit habitual responses, and make quick decisions in complex situations, which helps enhance their overall level of executive functions (*Diamond, 2013*). Therefore, if we wish to enhance children's executive functions during physical education teaching, we can combine these

two teaching methods with open motor skills and closed motor skills to achieve better intervention effects.

According to the subgroup analysis by exercise intensity, both moderate-intensity (ES = −0.64) and high-intensity exercise (ES = −0.63) significantly improved the inhibitory control function in children with overweight and obesity, with no significant difference between them ($P > 0.05$). When analyzing exercise intensity, it is necessary to mention the inverted U-shaped relationship between exercise intensity and brain arousal. Some studies have shown that the brain is at an optimal level of arousal during moderate-intensity exercise, which is more conducive to the allocation of cognitive resources. Therefore, moderate-intensity exercise has a significant effect on inhibitory control in children with overweight and obesity (*Zhang & Liu, 2019*). However, this study showed that high-intensity exercise also had a significant intervention effect on inhibitory control in children with overweight and obesity. The reason for this may be that for different types of cognitive tasks, the relationship between exercise intensity and changes in cognitive performance presents different trends. For relatively simple perceptual tasks, exercise intensity and cognitive performance were linearly related. Before reaching maximum oxygen uptake, the greater the exercise intensity, the better the cognitive performance (*Lambourne, Audiffren & Tomporowski, 2010*).

According to the subgroup analysis by age, exercise interventions significantly improved inhibitory control abilities in both children with overweight and obesity under 12 years old and those aged 12 and above. However, the intervention effects were more pronounced in children younger than 12 years old (ES = −0.69). The differences in the effects of exercise interventions may be due to variations in brain development and neurocognitive function across different age groups. Firstly, the critical period for the development of inhibitory control functions in children is between 3 and 7 years old (*Best & Miller, 2010*). During this stage, children's nervous systems develop rapidly, especially the prefrontal cortex, which is closely related to inhibitory control (*Best & Miller, 2010*). From 7 to 12 years old, inhibitory control functions continue to improve steadily (*Diamond & Lee, 2011*). After the age of 12, although inhibitory control functions are still maturing, the rate of development begins to slow down (*Luna et al., 2004*). Therefore, the different developmental stages of inhibitory control might be one of the reasons for the observed differences. Secondly, children under 12 years old have higher neural plasticity (*Johnson, 2001*). Exercise interventions have been proven to enhance neural plasticity and promote the release of brain-derived neurotrophic factor (BDNF), thereby improving inhibitory control functions (*Voss et al., 2011*). Finally, children aged 12 and above are gradually transitioning into puberty, experiencing significant physiological and psychological changes. Fluctuations in hormone levels may affect emotional and behavioral control, potentially weakening the effects of exercise interventions (*Blakemore, Burnett & Dahl, 2010*). In contrast, children under 12 may exhibit better compliance, leading to more effective outcomes from exercise interventions.

This study has the following limitations: (1) due to the relatively small number of studies on the impact of exercise interventions on the executive functions of children with overweight and obesity, the statistical power in conducting meta-analyses was somewhat

limited; (2) owing to the complexity of executive functions, different researchers in the included studies used various measurement methods, which may have introduced some variability into the results; (3) the evidence level for working memory, cognitive flexibility, and attention in this study is low, and the results may change as more evidence emerges; (4) this study only explored the impact of exercise interventions on the executive function of children with overweight and obesity, and did not deeply investigate how factors such as weight changes affect executive function.

## CONCLUSIONS

The results of this meta-analysis indicate that exercise interventions have significant benefits for inhibitory control, working memory, and cognitive flexibility in children with overweight and obesity, but the impact on attention is not significant. Moreover, the effects of inhibitory control interventions are influenced by exercise duration, exercise intensity, exercise type, and age.

### Funding
The authors received no funding for this work.

### Competing Interests
The authors declare there are no competing interests.

### Author Contributions

- Pengfei Wang conceived and designed the experiments, performed the experiments, analyzed the data, prepared figures and/or tables, authored or reviewed drafts of the article, and approved the final draft.
- Ying Meng conceived and designed the experiments, prepared figures and/or tables, authored or reviewed drafts of the article, and approved the final draft.
- Jinnian Tong performed the experiments, authored or reviewed drafts of the article, and approved the final draft.
- Tiance Jiang conceived and designed the experiments, performed the experiments, analyzed the data, prepared figures and/or tables, authored or reviewed drafts of the article, and approved the final draft.

### Data Availability
This is a systematic review/meta-analysis.

### Supplemental Information
Supplemental information for this article can be found online at http://dx.doi.org/10.7717/peerj.19273#supplemental-information.

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
