# Peer review of "Effects of exercise intervention on executive function in children with overweight and obesity: a systematic review and meta-analysis"

_PeerJ, doi:10.7717/peerj.19273_

## Round 0.1 · original submission · Major Revisions

Upon review of your manuscript the reviewers have identified several areas for improvement. In particular, queries around experimental design and the prospect of additional analyses (where relevant/applicable).

·

Basic reporting

The introduction and discussion needs to be updated for different mechanisms of change. Additional references need to be added relating to the various meta-analyses done to date in typically developing children.

Experimental design

Sub-group analysis by age if there are sufficient studies. Please also report the GRADE evidence profile of each outcome.

Validity of the findings

The discussion needs to be updated for other mechanisms of change.

Reviewer 2 ·

Basic reporting

Basic reporting/references/figures:
- Title: should be “in children with overweight and obesity”
- Please correct for grammatical errors throughout
- Please use person-first language (“child with obesity” not “obese child”)
- Background: I’m not sure it’s true that a decrease in executive function is a cause of obesity. Citations refer to an association only.
- Discussion section is quite long; the first paragraph re-iterates what was discussed in the intro; perhaps can shorten and tighten up the discussion to be more succinct and clear.

Experimental design

Experimental design:
- Systematic review and meta-analysis is a strong study design to answer the research question
- Please clarify if the use of the term “intervention” in the search will omit studies of exercise in children that is not part of an intervention (ie epidemiological studies of “real-world” or “naturally occurring” exercise in children, stratified by age)
- Surprised that there were only 13 studies in the inclusion; is it possible that some studies were missed given the specific exercise names given in the inclusion?

Validity of the findings

Validity of findings:
- Could there be confounders to the observed effects on cognition? Could weight loss alone have led to the improvement? Did this analysis account for that?

Additional comments

none

---

## Round 0.2 · accepted · Accept

Dear Authors,

Congratulations on the acceptance of your paper.

·

Basic reporting

All my questions have been addressed.

Experimental design

All my questions have been addressed.

Validity of the findings

All my questions have been addressed.

Additional comments

None.